# The Domestication of Machismo in Brazil: Motivations, Reflexivity, and Consonance of Religious Male Gender Roles

**DOI:** 10.3390/bs14020132

**Published:** 2024-02-12

**Authors:** H. J. François Dengah, William W. Dressler, Ana Falcão

**Affiliations:** 1Department of Sociology and Anthropology, Utah State University, Logan, UT 84322, USA; 2Department Anthropology, The University of Alabama, Tuscaloosa, AL 35487, USA; bill.dressler@ua.edu; 3Independent Researcher, Ribeirão Preto 14050-100, SP, Brazil; falcaoanni@gmail.com

**Keywords:** cultural models, cultural consonance, reflexivity, Brazil, gender roles

## Abstract

The relationship between culture and the individual is a central focus of social scientific research. This paper examines motivations that mediate between shared culture norms and individual actions. Inspired by the works of Leon Festinger and Melford Spiro, we posit that social network conformation (the perceived adherence of one’s social network with norms) and internalization of cultural norms (incorporation of cultural models with the self-schema) will differentially shape behavior (cultural consonance) depending on the domain and individual characteristics. For the domain of gender roles among Brazilian men, religious affiliation results in different configurations of the individual and culture. Our findings suggest that, due to changing and competing cultural models, religious men are compelled to reflexively “think” about what masculinity means to them, rather than subconsciously conform to social (hegemonic) expectations. This study demonstrates the importance of considering the impetus of culturally informed behaviors and, in doing so, provides a methodological means for measuring and interpreting such motivations, an important factor in the relationship between culture and the individual.

## 1. Introduction

The relationship between culture and the individual has been a central focus of social scientific research for over a century. Social structures and culture inform, guide, and constrain the thoughts and actions of individuals. Individuals, however, are also independent agents, employing actions that are not wholly determined by structural and cultural constraints. The tension between culture and agency, and the relationship of the individual and society, has been the central focus of many theoretical endeavors, including those of Durkheim, Bourdieu, and Giddens. Potential mediators that shape how culture and the individual interact have also been examined, with the anthropologist Melford Spiro proposing a process of “internalization”, psychologist Leon Festinger advancing “social comparison”, and sociologist Margaret Archer offering the notion of “reflexivity” to bridge the culture–individual divide. There has been, however, little methodological advancement in this area of inquiry. The methods and theories of cultural consensus analysis [1] and cultural consonance [2] provide approaches for linking individual behaviors with shared sociocultural processes, enabling the evaluation of possible mediators of the culture–individual relationship.

Research using these cognitive anthropological approaches demonstrates that humans’ actions are embedded in shared understandings. Social and cultural structures compel conformity via constructions of normalcy and deviancy; individuals who are more consonant with societal and cultural expectations possess greater social and cultural capital, are better positioned within the social hierarchy, and demonstrate better health outcomes [2]. These findings, replicated by a number of studies, nevertheless subtly privilege a structural vantage point in the relationship between the individual and culture. As a result, research may inadvertently overstate the notion that cultural norms or models are “self-motivating”, and individuals are compelled, somewhat automatically, to enact them [3]. 

The means and motives by which individuals are enabled and choose to enact culturally prescribed behaviors (or not) are no less important for understanding the individual–culture/structure relationship. Outside of contexts of doxa (or unexamined and self-evident cultural constructions), cultural models exist within pluralistic settings, with competing and overlapping ideals of orthodoxy and heterodoxy available for implementation. Human agents, by choice or by force, are motivated, coerced, and/or compelled to follow certain models over others within specific contexts. Such pressures to enact cultural models were articulated by Roy D’Andrade [3] and later developed by cognitive anthropologists such as Spiro, Quinn, Strauss, and others [4,5,6]. They argued that cultural norms are motivated into practice through a combination of external social forces and internalized ideals. Until recently, however, the field lacked a compelling methodology to fully examine how individuals variously react to, and are influenced by, cultural norms.

Utilizing cognitive anthropological methodologies, this paper evaluates the motivations by which gender norms are enacted in Brazil among religiously inclined men. In 1995, Elizabeth Brusco [7] published her influential ethnography *The Reformation of Machismo*, which asks “What happens to Colombian men who convert to Evangelical Christianity?” Among other things, Brusco argues that male gender roles are reoriented away from the male prestige complex of machismo and toward household and domestic (i.e., traditional female) goals. Like many anthropological works, Brusco relies on qualitative data gathered through careful participant observation and interviews to make her argument. We take Brusco’s findings as a starting point for our own research but contend that mixed-method and structured approaches (e.g., cultural consensus and cultural consonance) provide a valid empirical evaluation of qualitative and interpretive findings [8,9]. That is, we seek to not only replicate via quantitative methods Brusco’s findings in Brazil among a wider sample of religious men, but also examine the relationship of individuals and culture via the motivations that compel religious Brazilian men to enact alternative forms of masculinity. Specifically, we ask the following:

RQ1: are religious men expected to enact gender roles more closely aligned with feminine models?

H1a: the religious male model will be more closely aligned with terms associated with the Brazilian female model (compared with the secular Brazilian male model).H1b: the religious male model will eschew terms associated with negative aspects of machismo and endorse items associated with domestic and familial duties.

RQ2: what motivates men to enact these religious gendered behaviors?

H2a: men who are active in their religious communities will have greater consonance.H2b: men who internalize the model as personally meaningful will have greater consonance.

We organize our paper into the following sections: First, we briefly examine the dominant social constructions of gender in Brazil. The norms associated with the gender complexes of machismo and marianismo (idealized masculinity and femininity) provide the backdrop for analyzing the articulation between structure and individual action. Next, we provide background on the cognitive anthropological theory associated with cultural consonance and offer a means of evaluating culture–individual mediators via internalized (à la Spiro) and external/social (via Festinger) motivations. We then present results from a mixed-methods study of Brazilian gender roles. Results are then discussed with respect to theories of culture and the individual. 

### 1.1. Gender in Brazil

Gender is a useful domain for the evaluation of the interplay between culture and the individual, particularly in Brazil. Cross-culturally, gender is an essential form of identity display [10]. Social norms of gender, as such, become a primary marker of identity and a motivator of behavior. Brazilian anthropologist Roberto Da Matta [11] argues that Brazilian society, both ideologically and spatially, is bifurcated along such gender lines. The male orientation is denoted by the *ruas* (streets), *praças* (plazas), and *cidade* (city), as well as the actions that take place there (e.g., professional work, drinking). In contrast, female space is epitomized by the *casa* (house), designated as a site of reproduction and nurturing and as a place protected from the outside world. The gendered organization of Brazil is often framed by the machismo and marianismo archetypes. These gender roles exist both in opposition to one another but also in a symbiotic relationship [12]. While these gender models may be stereotypical, they are salient cultural constructs that shape the thoughts, perceptions, expressions, and actions of Brazilians [13].

Machismo is a traditional form of hypermasculinity that emphasizes male domination [14,15]. This gender role situates manliness in contrast to and even in opposition with femininity. Men are expected to be strong and assertive versus the submissiveness expected of women [16,17]. This expectation of dominance and authority extends to all the social and familial roles a man may occupy. For instance, men are expected to be the primary supporter and decision maker in the household, defending the safety and honor of the family (i.e., *caballerismo* or chivalry) [18]. Similarly, men are expected to show dominance with non-familiar actors, and limit emotional expression in order to avoid showing weakness. In these public spaces, such as the *ruas* (streets), *barzinhos* (bars), and *jogos de futebol* (especially neighborhood soccer games), men often gather with one another in these masculine third places (a community space separate from home and work), where they are socialized within these gender norms.

Within these all-male gatherings, one’s masculinity (and honor) is very much on trial in the eyes of one’s peers. In small talk, men usually avoid expressing honest opinions or feelings less they open themselves up for ridicule. Instead, talk usually involves verbally jousting with one another and using innuendos embedded in Portuguese to symbolically “feminize” and “penetrate” the other, thereby gaining status by dominating (i.e., feminizing) others [12]. When not verbally sparring, it is common for talk to turn towards sexual exploits, including stories of *putarias* (brothels), which are common in Brazil. Indeed, given this need to aggressively demonstrate manliness, it is perhaps unsurprising that violence, substance abuse, and the pursuit of sexual conquests are endemic. For example, Ribeirão Preto, the site of this study, has a male alcohol dependence rate of 43.5% [19]. And Brazil has been labeled one of the most dangerous places for women, with a high femicide rate (4.8 for every 100,000 people) and an estimated domestic violence event occurring every two minutes [20,21]. Such exaggerated displays of masculinity cross racial and class boundaries in Brazil. Indeed, former Brazilian President Jair Bolsonaro, who was equally celebrated and derided by his voters for his machismo behavior, is famous for downplaying the COVID-19 virus, telling the public to stop being “sissies” [22]; describing a rival female politician as “not worth raping; she is very ugly” [23]; making fun of the penis size of an Asian tourist [24]; and promoting foreign sex tourism with Brazilian women in an effort to discourage “gay tourism” [25].

Marianismo is the counterpart to machismo and refers to the cultural norms of femininity. The term “marianismo” refers to the Virgin Mary, who is the paragon of the ideal woman—virtuous, pure, self-sacrificing, and maternal [26,27]. In contrast to machismo, where the focus is on a man’s authority over others, the emphasis of marianismo is a woman’s service (and submissiveness) to others. Marianismo values, above all else, a woman’s self-sacrifice for the needs and wellbeing of her children and family. Her status comes from the suffering she has endured (at the hands of men, especially her husband) and the resulting devotion given to her by her children [12]. This focus on the family is both in opposition to but also an accommodation to the machismo expectations placed on men. Because machismo positions masculine spaces outside the house (and can encourages anti-familial behaviors), women often have to take on the role of guardians, nurturers, and providers of the family and household (contrary to *caballerismo*). The reality of gender roles for many women in Brazil means a contradictory existence. Women are expected to be virginal and pure but also mothers. They are expected to nurture the household, but absentee husbands require their employment outside the house. So, while marianismo does subordinate women in a patriarchal regime, it can also provide empowerment as the spiritual, moral, and (often) financial core of the household and community [26,28]. Even though machismo and marianismo may be dominant cultural models of gender, they are just two ways of performing gender out of many. Across Brazil, there are other ways to display and enact masculinity and femininity in addition to non-binary and transgender identities (e.g., *travesty*). One such alternative model of masculinity is that of the “religious man”.

Religions can provide alternative gender roles to that of dominant society. Faith-based communities, particularly those found in Western and secularized countries, are productive areas for the development of alternative and competing cultural norms. Religions frequently exist within a somewhat precarious position in the wider society—they need to provide an alternative style of life to set themselves apart from secular society while offering lifestyle improvements to potential converts. However, too much separation creates barriers for conversion and participation, limiting the faith’s growth and longevity [29]. As a primary indicator of identity, gender ideals are often a point of distinction for faiths, including in Brazil. While Brazil is nominally Catholic, it is also a rich religiously pluralistic society and counts Abrahamic, indigenous, and African-derived faiths as major religions. The particularities by which any of these faiths endorse a specific gender role will differ, but a common framing is of religious masculinity in contrast to secular machismo. Ethnographies show that traits such as marital fidelity, sobriety (not necessarily abstention), hardworking, caring father/spouse, and household provider are male traits endorsed by Brazilian faiths across the spectrum [30,31]. Indeed, rather than promoting a specific or unique type of religious gender performance, much of the core description of “good religious men” involves a rejection of some of the more detrimental aspects of the machismo model. Brusco [26], in her study of Colombian Evangelical masculinity, argues that the Evangelical faith distinguishes itself from dominant society by explicitly rejecting key aspects of the male prestige complex that is associated with familial strife, domestic violence, and other social ills. While perhaps not as explicit within sermons and doctrine, other Brazilian faiths make similar critiques of machismo and, in doing so, orient models of “the good man” in distinct opposition. The degree to which religious traditions endorse similar styles of masculinity is an empirical question that will be taken up below. For the purposes of this study, we will use cognitive anthropological approaches to examine whether there is a shared religious model of masculinity and evaluate how it compares to both secular masculinity models (i.e., machismo) and female gender roles (i.e., marianismo), and determine what motivates religious men to follow a faith-based masculinity norm over secular machismo.

### 1.2. Cultural Models and Motivations

A cognitive theory of culture is based on Goodenough’s [32] definition of culture as that which one must know to function adequately in a given society. Knowledge is structured within domains that are organized spheres of discourse or, more prosaically, topics that arise in mundane discussion. Knowledge within a domain is encoded in models, consisting of the elements (terms and phrases) that make up the domain, along with the semantic, functional, and causal associations understood to link and contrast among those elements. Cultural models enable us to (more-or-less) accurately interpret our environment, especially the actions of others, and to direct our own actions. Cultural models range from the commonplace (e.g., eating in a restaurant) to the sublime (e.g., religious devotion). The shared knowledge informs and directs our actions (e.g., reading a menu or reading from scripture) and coordinates social interaction.

Roy D’Andrade [3] posited that a vertical integration of cultural models is associated with their motivational potential. For example, gender identity—being a man or a woman—could be thought of as a more abstract model that in turn entails models involving specific social settings and practices, such as “going to the bar” or “wearing fashionable clothing” (see [33], and [34] for a discussion of *molar* models). Yet, models, in themselves, “do not automatically impart motivational force” [35] (p. 13). Rather, these models are motivating because they are culturally important in social situations and in individual’s psyches. Put differently, models are motivating due to the pressures of “socializing agents” for conformity; the psychological drive of models incorporated into the “self-schema”; the predictive or goal-seeking value they confer; and the degree to which a model is seen as “natural and right” (e.g., doxa, molar models) [36] (p. 227).

Of these four conditions, which need not be mutually exclusive, some domains may be more motivated by external social pressures or via internalized self-schemas [37]. Certain domains, due to the nature of the model, the social/cultural conditions within which they arise, and/or individual variation, will be motivated by external social pressures that reward conformity and punish deviance. Other domain actions arise from the internalized pressures from the psyche when models are incorporated within one’s world view and become part of the “self-schema of the individual” [36] (p. 227) (see also [4,38]).

Unfortunately, despite the apparent utility of these motivational theories, these ideas have not been widely incorporated into research on cultural models. Recent developments in theory and method have, however, provided researchers with the toolkit to examine adherence to cultural models and factors underlying that more precisely These include cultural consensus analysis and the concept and measurement of cultural consonance. Cultural consensus analysis is used to analyze the degree of sharing of knowledge in a particular cultural domain; if there is no sharing, there is no cultural model [1]. Based on the degree of sharing, cultural consensus analysis also provides a “cultural best estimate” of how a reasonably knowledgeable individual in that specific society would answer the set of questions, thus providing a prototype of shared norms. Cultural consonance is the degree to which individuals, in their own behaviors, approximate the prototypes for behavior encoded in cultural models. By matching reported behaviors to the prototypical answers regarding shared norms, a scale of cultural consonance can be derived, which in turn can serve as the dependent variable to determine which factors motivate cultural consonance [2].

Next, we proceed to a more detailed discussion of what those motivators of cultural consonance might be.

### 1.3. Social Comparison Theory

Leon Festinger’s [39] theory of social comparison examines how individuals employ social norms to assess and shape their own identity and behaviors. By evaluating one’s self in relation to others, individuals gain both cultural knowledge and self-knowledge. These comparisons are ubiquitous in daily interactions and often occur outside of conscious awareness [40]. This proclivity for evaluation allows us to determine our place within social hierarchies, formulate appropriate goals and behaviors, and reflect on one’s opinions and values in relation to known social actors. Festinger proposed that social comparisons are more prevalent during novel and/or stressful situations, where appropriate actions may be unclear. In these situations, comparisons reduce the stress and anxiety that occurs in unpredictable contexts by enabling individuals to gain information about appropriate actions through observation of their peers. Indeed, this is likely why social comparison is so prevalent during adolescence, as the many novel social interactions have potential costs to be avoided and self-knowledge is at a minimum [41,42].

Festinger identifies two cognitive processes that can occur during social comparison. Individuals may assimilate by bringing their own behaviors and opinions in line with that of the social group, minimizing any perceived differences. This can occur in a variety of contexts, but particularly when the individual feels that they are deviant with the overwhelming consensus of the group. For example, the Asch conformity line experiments demonstrate that individuals will override their own sense of reality in order to conform to the shared opinions of the group [43]. The other process involves contrasting comparisons, where individuals emphasize the differences between themselves and other social actors. This is more prominent in situations where individuals are comparing themselves to those considered inferior (e.g., lower SES than themselves) or to out-group individuals (e.g., a religious person comparing themselves to a non-religious individual).

A key aspect of this theory is that the evaluations are subjective. They are based on what the individual perceives to be the shared values and actions of the comparative social group and their own related positionality. That is, it matters less what the actual objective opinions of each individual in the group are, rather what the individual perceives them to be. Methodologically, this means that every individual in a person’s social network need not be evaluated for their adherence (or consonance) with a particular social norm. Instead, the subjective appraisal of what someone understands their peer group’s norms to be is more insightful for social comparison.

### 1.4. Internalization

Melford Spiro’s [4] theory of internalization seeks to explain how cultural knowledge is learned and how it shapes behaviors. In essence, cultural knowledge is transmitted via socialization and enculturation from something external to the individual to something that is within the cognitive and motivational structures of the person. Spiro identified four distinct stages in the process of internalization. First, an individual is exposed to a cultural norm. Following this, that particular piece of cultural knowledge is transmitted to the individual in that it is learned but not acted upon. In the third stage, the cultural norm is incorporated or internalized into one’s cognitive and behavioral repertoire. Finally, in the fourth stage of internalization, the cultural norm becomes an identifying aspect of one’s self-concept and is an organizing and motivating force.

Spiro’s internalization theory, however, has been critiqued for its metaphor of “depth” and for not applying to all types of knowledge [44]. And indeed, Spiro’s model does appear to privilege knowledge learned through socialization compared with that attained through unsupervised enculturation processes (such as embodied knowledge, which may skip stages 1–2). These critiques notwithstanding, the typology does offer a useful spectrum for evaluating how some types of knowledge can become more *integrated* within one’s cognition. One’s concept of gender can be a master organizing force, such as in societies like Brazil with a clear gendered bifurcation of space, or when it is integrated with religion, as in the case below.

Dengah, Falcão, and Henderson [45] drew on these theoretical orientations to determine whether there were shared cultural models of gender that corresponded to the distinction between machismo and marianismo, and to identify the strongest correlates of cultural consonance with each cultural model. They found that, indeed, there were consensus models identifying a *machista* male role and a *marianista* female role. For women, cultural consonance with the female role was most strongly associated with internalization of that role, while for men cultural consonance with the male role was most strongly associated with the degree to which men perceived their immediate social network as consonant with that role.

At this point, we turn to the question of the degree to which religiously oriented men adhere to novel cultural norms provided by religion for male gender roles in Brazil, and to examine how social comparison processes and internalization are motivating forces in that adherence.

## 2. Methods

Data were collected in 2018 and 2019 in Ribeirão Preto, Brazil, a city of approximately 700,000 people in the state of São Paulo. The city was founded on coffee cultivation, and agriculture is still a foundation of the economy, although in recent decades it has grown as a regional financial, educational, and healthcare center.

Data collection proceeded in two stages. The first stage was devoted to eliciting the terms and phrases that Brazilians use to talk about the cultural domain of gender roles. Convenience sampling was used, recruiting respondents from public locations such as shopping malls, parks, and the city center. Within this sampling strategy, we also used a quota sampling frame. Approximately equal numbers of men (*n* = 24) and women (*n* = 23) were recruited and the average age (36.7 ± 13.8 years old) and income (3.4 ± 2.1 minimum salaries or about BRL 1000/month) approximated that of the city as a whole, although the sample has a somewhat higher educational status (with 70% having at least a high school education and 28% completing education beyond high school). Semi-structured interviews were conducted in which respondents were asked to describe gender roles endorsed in Brazilian society and in their faith-based communities (if relevant). Interviews were conducted until data saturation was achieved (i.e., respondents were no longer generating new terms to describe Brazilian gender roles).

Grounded theory text analysis (using the software Dedoose Version 9.0.17) was conducted to identify salient terms describing male and female gender roles [46]. Initial coding was conducted by the first author. The third author (a native Brazilian) then verified codes. As seen in other work (e.g., [11]), these respondents clearly contrasted the differences between male and female gender roles. 

For the second stage of the research, a separate sample of 60 women and 42 men was recruited, with an emphasis placed again on approximating age and income composition of the community (34 ± 11.4 years old; 3.9 ± 3.8 minimum salaries). A total of 75 individuals identified as belonging to a religion, 30 of whom were men (see Figure 1). Among the male sample, thirteen individuals identified as being part of the Protestant or Evangelical faiths, eight were Catholic, five practiced Espiritismo, two informants self-identified as Buddhist, and two practiced Afro-Brazilian faiths. Evangelicals were oversampled to examine Brusco’s hypothesis that evangelical men consciously position themselves in contrast to and reject key aspects of secular Brazilian masculinity (i.e., machismo). In the sample of 30 religious men, the average age was 36.6 (±12.6, range 21–67 years old), who earned 4.7 minimum salaries (±4.4) with 83% of our sample having some college or vocational education. Respondents were again recruited from public locations and identified as cis gender. While this was a modest sample, it was employed to test a focused hypothesis, using measures that were both theoretically and ethnographically precise, and incorporated both quantitative and qualitative data, thus strengthening the confidence in the results [47].

Cultural Consensus Analysis: Cultural consensus analysis (CCA) is an analytic model for testing the degree of sharing of knowledge among a set of respondents. Although not a factor analysis per se, CCA uses factor analytic methods to determine (a) the overall degree of sharing of knowledge; (b) the degree to which individual respondents agree with the aggregate; and (c) a consensus set of responses to the questions that represents the culturally best estimate of how a reasonably knowledgeable member of that society would answer the questions [1].

In this study, the terms and phrases describing normative gender roles elicited from the semi-structured interviews described above were formulated as questions regarding the importance of each trait to men or women. Terms of course applied to either men or women, and both genders responded to all questions, providing both comprehensive and gender-specific data to test for cultural consensus. Subjects responded “true” or “false” regarding the importance of a specific trait to specific genders. Also, respondents were prompted to think in terms of a trait’s importance as viewed in general in Brazilian society to emphasize the collective representation of gender roles.

Cultural Consonance: Cultural consonance is the degree to which individuals approximate behavior defined as normative in the cultural consensus analysis. To measure cultural consonance, each respondent was asked if they engaged or not in the behavior described by each item. For each individual, the items that were defined as important (i.e., “true”) for gender role behavior in the consensus answers derived from CCA were summed. This score thus represents the degree to which individuals reported actually conducting what the cultural consensus analysis identifies as culturally appropriate in terms of role behavior, or the degree to which each individual matched behavior regarded as culturally important. (Note that some items were regarded as inappropriate for males or females; these were reverse coded and included in each scale appropriately.)

Social Network Conformation: This scale measures the degree to which each individual sees their immediate social network as culturally consonant with each item assessing role behavior. Specifically, for each item, respondents were asked “Does the majority of your family and friends think this characteristic is important to being a woman (or a man)?” Items were summed to measure the degree to which an individual perceived their social network as conforming to the consensus model.

Internalization of Cultural Norms: To measure the degree to which individuals have *personally* adopted specific cultural gender norms, each respondent was asked if each trait was important in defining appropriate gender roles. When summed across each trait, this measure assessed the internalization of the consensus cultural models. 

The secular male (machismo) variables had a potential range of 0–22; the female (marianismo) model had a range of 0–28. The religious male variables had a possible range of 0–27. Variables were standardized for statistical analysis.

Covariates: Covariates included age (in years), socioeconomic status (SES), religious affiliation, and religious activity. SES was assessed as the first principal component of total household income, head of household income, and respondent education (the first principal component accounted for 69% of the variation). Respondents self-identified as having a religious affiliation or not. Individuals who were religious then disclosed the specific faith-based tradition they identified with. Members of Protestant faiths (e.g., Pentecostals, Evangelicals, Baptists) were oversampled due to other research aims and thus controlled for in this present study. Religious activity was measured by self-reported average frequency of attendance per week; informants were grouped by those who attended religious activities less than once a week, once a week, and more than once a week.

## 3. Results

The cultural consensus analysis was based on a mixed-gender sample of 75 self-described religious respondents, as both men and women jointly participated in the construction of gender norms. However, correlations and least squares regression focused solely on religious men (*n* = 30), whose descriptive statistics are presented in Table 1. 

Consensus was found for the religious male model with an eigenvalue ratio of 6.5, average competency of 0.64, and a single negative score (−0.07). Importantly, there were no differences in religious affiliation, supporting the notion that religions similarly distinguish religious masculinity apart from secular machismo, at least for the terms used in this model.

Figure 1 displays a Venn diagram depicting the composition of each model. As described earlier [45], for men, an overall male model was defined by chivalrous protection of the family (e.g., *provider*, *father*, *worker*), traits associated with the *ruas* (e.g., terms associated with going to *bars* and *drinking*), and female conquest (e.g., *promiscuous*). Also, as described earlier, the female gender role model corresponded to the pan-Latin American marianismo model (e.g., *mother*, *strong faith*, *faithful spouse*, *raising children*, *submissive*, and *suffering*). At the same time, there were modern and “progressive” qualities to the model valuing *education*, *independence*, and *working*.

A key finding for the current study was that religion reforms, even domesticates, machismo. The findings suggest that the religious male model is indeed more aligned with female traits compared with the secular male model (supporting H1a). The religious male model included all the familial traits associated with women and rejected key aspects of the machismo model, including excessive alcohol consumption and promiscuous behaviors (supporting H1b). Importantly, the model maintained an overall patriarchal focus, situating men as household heads, leaders, and providers. So, while the religious male model was aligned with female pursuits, it was anything but egalitarian (see [7]).

Again, it should be noted that agreement for this model was found across all religious informants, with no significant variation among the first or second factor for members of different faiths. This may be because the terms in this domain are rather general and do not focus narrowly on specific religious activities or rituals or more esoteric values promoted by certain denominations—such as the prohibition of soccer by some conservative Evangelical congregations. Further, religions define themselves apart from secular society by reorienting behaviors away from worldly pursuits and towards spiritual and family oriented goals. Within Brazil, religions create cultural separation through reorienting male behaviors away from the destructive prestige complex of machismo.

Finding evidence of shared cultural models, we can ask (RQ2) “What motivates men to enact these religious gendered behaviors?” To answer this question, we first look at correlations between our predictor variables and religious consonance (see Table 2).

We include the variable of Protestant identification, since Evangelical groups were oversampled. Lower socioeconomic status is associated with an increase likelihood of Protestant affiliation, supporting the ethnographic accounts and census data that portray Evangelical faiths as particularly popular among Brazil’s lower and lower-middle classes [48,49]. Being Protestant is associated with a greater social network endorsement of the religious male model and greater internalization of this model as a personal ideal, but is not significantly associated with consonance. However, being religiously active is predictive of internal motivation, as well as greater consonance with the model, supporting H2a. As expected, both motivators are significantly correlated with consonance. 

To understand the relationship more fully between individuals and shared cultural norms, we regress our behavior (consonance) variable on the motivation variables (social network conformation and internalization) (Table 3). 

The least squares linear regression shows that, while controlling for covariates (including religious affiliation and activity), internalization is predictive of male behaviors, explaining the bulk of the variance (and supporting H2b). That is, religious Brazilian men in this sample were more likely to enact the religious style of masculinity if they internalized the model as personally meaningful. 

## 4. Discussion

The analysis of how religious Brazilian men are motivated to perform a masculinity counternarrative sheds light on how individuals are differentially motivated to enact culture. The data show that, depending on the population and the domain, cultural performances (i.e., consonance) are compelled by different pressures. Our previous research indicated that secular Brazilian male identity performance (i.e., machismo) is motivated by one’s peers and social environment [45]. Controlling for age and SES, Brazilian men in general are more likely to enact the machismo role if they have greater knowledge of the gender norms (competence) (Beta = 0.33, *p* < 0.05) and view their social network as endorsing machismo masculinity (social network conformation) (Beta = 0.32, *p* < 0.05, R^2^ = 0.4). This *machista* male configuration of performing the gendered role is socially located in the street, bars, and fields on which football is played, spaces that are male and public [11]. The social pressure of correct gender performance compels compliance even if their own views differ, or else individuals leave themselves open for ridicule and social sanctions, often in the form of their manhood being challenged (see [10]).

For Brazilian women, the cultural consensus analysis clearly indicates that the *marianista* model is shared. Our previous research shows that women are more likely to be culturally consonant with that model if they internalize it (i.e., regard that configuration as personally important to them (Beta = 0.45, *p* < 0.01, R^2^ = 0.43); [45]). Brazilian women, according to this model, are expected to be loving and morally pure, and whose devotion and ability to suffer for her family knows no bounds. As described by DaMatta [11], the model is enacted in the domestic space of the home.

Religious Brazilian men, however, have a different model of masculinity and a different motivation for performing these cultural prescribed behaviors. Coinciding with Brusco’s description of religious men in Colombia, the ideal religious man in Brazil is more aligned with female domestic goals and rejects the deleterious aspects of machismo. This model still places men in positions of authority, though within the context of faith and family. Indeed, this gender model promotes a counter narrative of masculinity that is, quite explicitly, a rejection of a machismo prestige complex. Brazilian religions, to distinguish themselves from the “secular” world and to heal the (social) ills of their members, promote a style of masculinity that, while not challenging the patriarchal structure of family and society, promotes a “domesticated” style of masculinity that aligns men and women in family and domestic goals. Evangelical and Catholic sermons often warn men against the dangers of (excessive) drinking and extra-marital sexual liaisons, and Candomblé *terreiros* elevate the role of women and align male pursuits with the desires of the *mae-de-Santos* and the *orixa* goddesses.

In fact, some faiths make efforts to separate men from the third place that is the epicenter of Brazilian masculine performance—the *futebol* field. Evangelical faiths, such as the Assembleia de Deus, forbid their members from playing *futebol*, not because the sport is inherently evil, rather it is the peer pressure of the other men that can persuade an otherwise outstanding Christian to drink, smoke, and engage in elicit sexual encounters [48,50]. As one of our respondents, Jorge (24), explains:


*“To be an evangelical, you have to have a different (gender) attitude. Because sometimes you think, ‘I’m a man, I should be doing that!’, but as an Evangelical you can’t. But then you have friends who…after a game are at a bar or churrasco…and you hang out with them. But even if you weren’t intending to, you start to drink because of the influence of your friends, and end up getting mixed up in the wrong things”.*


Jorge is single, has a primary school education, and works as a barber. Making only minimum wage, Jorge is among the lower class of Brazil. His social status should make him likely to exhibit machismo traits, as hypermasculine performance is often viewed as compensation for a sense of inferiority [12,51]. By rejecting key aspects of machismo, these religious males realign and internalize their gender performance similar to marianismo women. Moreover, this internalization of culture as a personal belief occurs within the context of religiously framed gender roles. For men, adopting a religious role results in a rupture with traditional secular male performances and, with it, traditional ways of male bonding. For converts and the newly religious, this may result in the dissolution of friendships and social networks; even the life-long faithful face reduced social capital opportunities by eschewing and avoiding male-centric spaces of *barzinhos*, *putarias*, and *clubes*. Indeed, it is not uncommon for men who have rejected certain aspects of machismo to face ridicule, such as the derogatory term “*pau-mandado*” (henpecked, whipped). Being consonant with this religious male model requires substantial internalization of the model as a personal guiding principle for individuals. This model functions as a counter narrative of masculinity and, as such, is subject to marginalization and sanctions from dominant cultural and social institutions that operate to promote an orthodoxic model of machismo.

This tension that religious Brazilian men feel in relation to wider society regarding gender roles can be seen in the quotes of some of our informants. Lucas, 39, has a high school education and works as a construction worker. He is also a long-time evangelical who defines the religious man in contrast to the secular Brazilian man:


*“In my religion, the man, he was born to be the leader, the head of the house. This is biblical, he is the priest of the house…But I think, as an evangelical man, he has to be a person of integrity, honesty, who values the family…Some people think that a man wasn’t born to be inside the house, washing clothes, washing dishes, changing a baby’s diaper. But for me, I have no problem with that. I clean the house, wash clothes, I help my wife…But if I talk about this to other men, they’ll say I’m not being a man”.*


Lucas, who is married with children, defines his sense of masculinity in domestic terms. In doing so, he echoes Brazilian anthropologist Roberto da Matta’s description of the bifurcation of space in Brazilian society. Lucas views Brazilian society as not valuing the role of men in the house, even to the point of challenging his very gender identity. Yet, the ideal religious man for him *is* defined by his place in the house. For Lucas, his gender role comes from not only being the patriarchal leader of the household but also the helpmate of his wife, focusing on the welfare of his family and household. Nevertheless, his understanding of his masculinity is well-articulated and thought-out, suggesting a high degree of internalization. Indeed, given that some of his peers question his gender due to his actions, his motivation to perform his gender role comes from strong personal convictions rather than the expectations of society.

Jefferson is single, 34, and a practicing Catholic. Like Lucas, Jefferson feels like Brazilian society’s gendering of space makes it difficult for men to work at home or be homemakers.


*People talk about how women suffer from prejudice. But men also suffer. For example, a man must have a traditional profession. If the guy is a writer, he is labeled a bum. If he does things for the home, he is labeled a bum. A “man” (according to Brazilian society) is someone who leaves in the morning for work, has lunch, and comes back home to his family. That is what society deems traditional.*


It should be noted that Jefferson himself is employed as a salesman outside the house. He is also quick to explain that he, and society, feel that men should be the primary provider, but “women can work, that’s okay” as long as the man is the “final boss.” Yet, Jefferson is working through what he feels is an unfair judgement for men who take on non-traditional employment or participate in more things around the house. He still has a very patriarchal world view, endorsed by both religious and secular Brazilian models of masculinity, but nevertheless feels prejudice from what he sees as a limiting secular model of appropriate maleness.

Jorge, Jefferson, Lucas, and the other religious Brazilian men in this study frame their own understandings of gender in contrast with, and even opposition to, key aspects of secular Brazilian masculinity. Even areas of general overlap with the machismo role, such as the expectation of men to be the *chefe de casa* (head of the household), are still qualified as being different than the secular role by being explicitly in service of the wellbeing of the family and household. That is, for religious men, they must position themselves as contrary, even deviant, with widely shared norms of masculinity, and internalize their religious counter narratives to successfully perform as men. 

A more agent-led theory of cognition and behavior provides context and explanatory power to these data. D’Andrade [36] (p. 226) posits that “as models become more deeply internalized, they tend to include more functions.” Such “master motive models” are foundational within the self-schema and serve as a catalyst for employing related models and schemas. However, such master-level models are often abstract, molar, and strongly associated with other (equally complex) models. In these cases, common knowledge of a model may result in different interpretations and associated behaviors among individuals, based on the pattern of cognitive linkages [52]. Such “connectionist” models are thought to be formed via a combination of both shared conceptual knowledge and individual life experiences (c.f. Vygotsky’s mediating devices [53]). The differences in association can result in individuals sharing a common schema but acting on it in divergent ways.

For religious Brazilian men, they share the same knowledge of machismo gender roles as the rest of society. Yet, they display significantly different choices of gendered behavior. The data show that these men internalize and incorporate their religious identity as an important part of their self-schema. The reasons for this internalization, however, are varied. This study can only infer the underlying individual psychological and personal experiences that compel the men in this study to value and display gender the way they do. However, a decade of research among Brazilian Evangelical communities in the study area by the first author has demonstrated that men engage in religious activity for a myriad of reasons [50]. Some do so because they were enculturated and socialized into the religious culture at an early age. Others convert via the actions of a loved one or through a powerfully traumatic experience that compels them to reexamine their own self-schema. Each of these unique biographies creates particular connectionist networks, informing how and why these models are internalized and acted upon.

Importantly, the connectionism of models and the internalization of them is a continual process. Spiro [4] provides a description of the various levels of internalization—each successive level likely to have greater emotional and motivational power [3]. However, he does not articulate exactly *how* this internalization takes place. Sociologist Margaret Archer’s theory of reflexivity may be insightful here. Archer [54] (p. 4) defines reflexivity as “the regular exercise of the mental ability, shared by all normal people, to consider themselves in relation to their (social) contexts and vice versa”. At its core is the supposition that humans are not passive players in their social or cultural structures. On the contrary, Archer’s theory contends that (outside contexts of doxa) individuals consciously meditate, to one degree or another, on their relationship with these external forces. Echoing Spiro’s own conception of internalization, reflexivity is the ability to arbitrate on what society and culture expects from them, and then react accordingly. In other words, the reaction or behavior of individuals to culture (e.g., cultural consonance) depends on how individuals interpret and relate to it.

Archer defines four (mutually inclusive) modes of reflexivity as follows: *communicative* (derived from interactions and confirmation from social peers), *autonomous* (inner conversations that require no validation or conformation from others), *meta* (self-monitoring and critical inner conversations, often characterized by doubt and stress), and *fractured* (disorienting and distressing inner conversations that lead to inaction or inappropriate action) [55]. We see some of these types of reflexivity taking place in the ways in which religious Brazilian men understand their own gender roles. Jorge, Jefferson, and Lucas describe their own positionality and relationship with both secular and religious forms of masculinity in their own understanding of gender. Jorge’s reflection on gender potential displays both communicative and meta forms of reflexivity. He articulates that his thoughts and behaviors are influenceable, if not influenced, by his friends, which runs counter to his own personal beliefs of what a good evangelical man should be. With a pinch of pain on his face, Jorge implies that he too had become “mixed up in the wrong things”.

While Lucas’ articulation is somewhat more cogent than Jefferson’s, there is nevertheless a clear mediation of how each feel in relation to dominant cultural expectations of men. (Archer would suggest that the differences in articulation may point to different types of reflexivity, possibly a more autonomous form for Lucas and meta-style reflexivity for Jefferson). The quantitative data show a similar pattern. Those who internalize and incorporate a religious understanding of gender are more likely to enact corresponding behaviors in their own life. Indeed, for Archer, a key aspect of reflexivity is both the subjective interpretation of cultural structures and, equally importantly, how that informs and guides individuals’ actions in the world. Archer [56] contends that, as societies undergo cultural shifts (a process she calls morphogenesis), the need for reflexivity rises as previous “taken-for-granted” forms of practice (i.e., *habitus*) are contested. The historical transformation of religion brought about by Protestantism, as well as the consciousness-raising of feminist movements, has meant that the implicit knowledge provided by conventional cultural models is at least incomplete if not problematic. In other words, for Brazilian women and religious men, the doxic nature of gender roles is no longer tacit and requires them to actively “think about who they are…and modify their identity in the course of everyday being” [57] (p. 608). For traditional Brazilian men, however, no such pressure exists; their taken-for-granted gender role requires less reflexivity and thus the internalization of the cultural model is less of an influence.

There are some limitations to this study that can inform future research. The sample size of this study skewed a more educated population than the general population. Archer [58] argues that, while reflexivity may vary depending on context, individuals tend to display certain reflexivity styles depending on their own autobiography. It is reasonable to suspect that education provides individuals with an increased capacity for (internally) vocalizing their own subjectivity and positionality with regard to culturally constructed norms and values. Along these lines, a parallel linguistic analysis would be beneficial methodologically and theoretically. Finally, this study is limited by a small sample size. Given the large amount of variance explained by the internalization variable, this is unlikely a spurious association, but a larger (educationally) diverse sample would provide a better assessment of how religious men relate to and are motivated by culture.

## 5. Conclusions

The religious marketplace of Latin America, particularly Brazil, is rapidly changing. In what has been called the most Catholic country in the world, Evangelicals now make up a third of the population and are more active in the pews than the largely nominal Catholic majority. The implications of this shift are far-reaching. From politics to television to social media, these faiths are actively challenging and remaking long standing Brazilian norms and institutions. In doing so, gender roles are simultaneously being affirmed and reformed. The traditional marianismo role of women is reinforced by these conservative faiths, emphasizing their role as wives and mothers. The male role, however, is rearranged—rejecting parts of the machismo model that are detrimental to the family and incorporating female aspects to align male goals with the domestic sphere. This realignment of religious masculinity has implications not only for society but for how individuals relate to society.

For the domain of gender roles among a Brazilian sample, sex and religious affiliations (for men) result in different configurations of the individual and culture. Findings suggest that, due to changing and competing cultural models, Brazilian women and religious men are forced to reflexively “think” about which gender role to follow. This meditation is necessary for them to successfully navigate through cultural space, compared with non-religious Brazilian men, whose gender role is so omnipresent and ubiquitous that it requires less negotiation of meaning between the individual and the shared model.

The ways in which culture is motivated into behavior can vary. Certainly, specific cultural domains and individual biographies can shape unique conditions for enactment. However, as this study shows, motivations can also be particularly patterned by subgroups as well. The common experiences of individuals position them in similar locations vis-à-vis the cultural environment. In this case, the motivation to take up alternative (heterodoxic) cultural behaviors requires an internalized locus of inclination in order to resist the habitus and social pressures of more dominant cultural norms.

The inquiry of how individuals are motivated to perform different behaviors is certainly not new. What has been lacking, however, is a clear methodological way of identifying a specific cultural domain and then turning this description of culture into a measure of motivation and behavior. Using the cultural consonance approach, this study demonstrates a methodological means for measuring and interpreting motivations, an important mediator in the relationship between culture and the individual.

## Figures and Tables

**Figure 1 behavsci-14-00132-f001:**
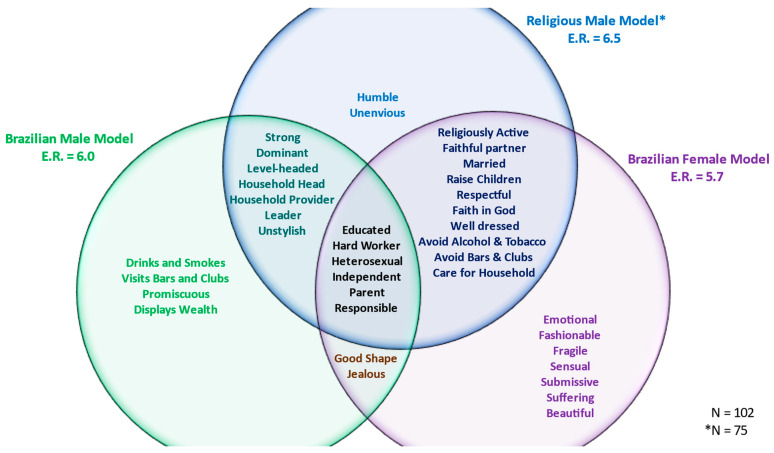
Comparison of gender models.

**Table 1 behavsci-14-00132-t001:** Descriptive statistics of key variables for religious men.

Column 1	Range	Mean (Standard Deviation)
SES	−1.5–3.2	0.2 (± 1.1)
Age	21–67	36.6 (± 12.6)
Religious Affiliation (Protestant)	43% Protestant	
Religious Activity	<1/week = 6; 1/week = 15; >1/week = 9	
Competence	−0.7–0.9	0.6 (± 0.3)
Social Network Conformation	8–27	18.2 (± 5.6)
Internalization	8–27	21.6 (± 4.2)
Cultural Consonance	10–27	21.6 (± 4.3)

**Table 2 behavsci-14-00132-t002:** Correlations of research variables.

	Age	Religious Affiliation	Religiously Active	Comp.	Social Conf.	Internalization	Consonance
SES	−0.11	−0.36 *	−0.13	−0.21	−0.08	−0.01	0.05
Age		−0.37 *	0.18	0.14	−0.01	0.13	0.22
Religious Affiliation (Protestant)			0.45 **	0.08	0.38 *	0.41 *	0.34
Religiously Active				−0.20	0.25	0.37 *	0.57 **
Competence					0.42 *	0.29	0.11
Social Network Conformation						0.67 **	0.57 **
Internalization							0.83 **

* Correlation is significant at the 0.05 level. ** Correlation is significant at the 0.01 level.

**Table 3 behavsci-14-00132-t003:** Regression models of male religious cultural consonance (R^2^ = 0.79).

Column 1	Religious Male Sample
	Standardized Coefficients (Beta)
SES	0.09
Age	0.07
Religious Affiliation (Protestant)	−0.05
Religious Activity	0.30 *
Competence	−0.05
Social Network Conformation	0.07
Internalization	0.70 **

* Correlation is significant at the 0.05 level. ** Correlation is significant at the 0.01 level.

## Data Availability

The raw data supporting the conclusions of this article will be made available by the authors on request.

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
