# Peer review of "The Domestication of Machismo in Brazil: Motivations, Reflexivity, and Consonance of Religious Male Gender Roles"

_behavsci, 2024, doi:10.3390/bs14020132_

Round 1
Reviewer 1 Report
Comments and Suggestions for Authors
The article facesan interesting and relevant topic, but some methodological choices are debatable.
In particular the analysis focus only on two sub-samples of 75 and 27 cases and, despite the limited size of these samples, not only aims at comparing the two sub-samples but even adopts a quantitative approach of analysis (moreover on the basis of quantitve indexes which is not clear how have been built). The authors even develop a regression model with 7 variables on 75 cases, and use indexes of statistical significance on this small and non statistically representative sample.
The quantitative part of the article is then unacceptable and must be eliminated.
Finally, conclusions are very short and then unsatisfying: the authors have 6 pages of theoretical reflection, so they must propose a wider conclusion where they discuss their empirical result in a dialogue with these reflection.
Reviewer 2 Report
Comments and Suggestions for Authors
l. 24-100: Your introduction is compelling: you do a good job of surveying both the field in general and Brazilian masculinity in particular, situating your research questions and your methodology in a clear and understandable way. Nice work.
l. 116: missing ")"
l. 153: Appreciate the opposed to/accommodation to take
l. 176: Good transition to religion
Sect. 1.1: Really clear, well-researched, and compelling analysis.
l. 218-229: I appreciate the nuanced way of presenting the distinction between model and motivation.
Sect 1.2: You do an excellent job of framing your methodological approach, defining what it is, why it hasn't been prominent, and how you make it work.
l. 270-280: I really appreciate your slow, thoughtful approach to identity formation.
l. 287-289: Your claim here is quite believable at this point. Nice work.
l. 291-300: Great breakdown of Spiro.
l. 301-347: You do a thorough job of describing your choices in terms of obtaining a representative sample.
Overall, Method section was both clear and illuminating, advancing your research by educating your reader on thoughtful distinctions that reinforce earlier work and take it a step forward.
Page 9: I appreciated the two ways of displaying data.
l. 466-469: Another really nice, subtle distinction.
493-506: The payoff of your work above becomes quite clear in the discussion here.
l. 493-566: Great integration of specific comments.
l. 575-586: Again, I really appreciate you pausing to work through more nuanced and complex theoretical notions to ensure your reader understands the care with with you make distinctions.
l. 611-614: I liked how you presented reflexivity with a nod back to Spiro, helping to advance your reader's understanding from a learned standpoint.
l. 629-647: Weaving reflexivity into a more complex understanding of a culture in change is well done.
Overall: This is a fantastic, excellent contribution--well written, well designed, and nicely articulated. Great work!
Reviewer 3 Report
Comments and Suggestions for Authors
The issue of how men's conception of idealized gender roles can change as they become more religious is an important one. There are some places where you might expand and restructure to strengthen the argument.
1. You can move the first paragraphs discussing the relationship between the individual and culture to the section where theoretical matters are discussed. This would allow you to move up the statement describing what this article is about, and then move into gender roles in Braziil.
2. Please correct the spelling of Columbia and Columbian to read Colombia and Colombian throughout.
3. In addition to Brusco, you can discuss Evelyn Steven's and Elizabeth Jelin's definitions of conceptions of machismo and Marianismo, and pan-Latin American constructions of gender role ideals respectively.
You can include a chart that provides more information about the participants' who you discuss individually -- to provide a sense of their ages, marital and paternal status, education level, perhaps occupation. This will strengthen your analysis of their interview data as it contextualizes them better.
Your conclusion can be elaborated to position your findings within a larger discussion of the changes that come with increased religiosity in Latin America and more generally.
Comments on the Quality of English Language
Overall, this is a solid start but the authors can reorganize as suggested above, and better better contextualize the Brazilian case within a broader framework that discusses other locales within Latin American and questions of religiosity more generally.
